# Liver Fibrosis Is Enhanced by a Higher Egg Burden in Younger Mice Infected with *S. mansoni*

**DOI:** 10.3390/cells13191643

**Published:** 2024-10-02

**Authors:** Heike Müller, Jan K. Straßmann, Anne S. Baier, Verena von Bülow, Frederik Stettler, Maximilian J. Hagen, Fabian P. Schmidt, Annette Tschuschner, Andreas R. Schmid, Daniel Zahner, Kernt Köhler, Jörn Pons-Kühnemann, Daniel Leufkens, Dieter Glebe, Surmeet Kaur, Max F. Möscheid, Simone Haeberlein, Christoph G. Grevelding, Ralf Weiskirchen, Mohamed El-Kassas, Khaled Zalata, Elke Roeb, Martin Roderfeld

**Affiliations:** 1Department of Gastroenterology, Justus Liebig University, 35392 Giessen, Germany; heike.mueller@innere.med.uni-giessen.de (H.M.); jan.k.strassmann@med.uni-giessen.de (J.K.S.); anne.s.baier@med.uni-giessen.de (A.S.B.); verena.von-buelow@innere.med.uni-giessen.de (V.v.B.); frederik.stettler@innere.med.uni-giessen.de (F.S.); maximilian.j.hagen@med.uni-giessen.de (M.J.H.); fabian.p.schmidt@med.uni-giessen.de (F.P.S.); annette.tschuschner@innere.med.uni-giessen.de (A.T.); elke.roeb@innere.med.uni-giessen.de (E.R.); 2Department of Internal Medicine III, Justus Liebig University, 35392 Giessen, Germany; andreas.schmid@innere.med.uni-giessen.de; 3Central Laboratory Animal Facility, Justus Liebig University, 35392 Giessen, Germany; daniel.zahner@zvth.uni-giessen.de; 4Institute of Veterinary Pathology, Justus Liebig University, 35392, Germany; kernt.koehler@vetmed.uni-giessen.de; 5Institute of Medical Informatics, Justus Liebig University, 35392, Germany; joern.pons@informatik.med.uni-giessen.de (J.P.-K.); daniel.leufkens@informatik.med.uni-giessen.de (D.L.); 6Institute of Medical Virology, National Reference Center for Hepatitis B Viruses and Hepatitis D Viruses, German Center for Infection Research (DZIF; Partner Site Giessen-Marburg-Langen), Justus Liebig University, 35392 Giessen, Germany; dieter.glebe@viro.med.uni-giessen.de (D.G.); surmeet.kaur@viro.med.uni-giessen.de (S.K.); 7Institute of Parasitology, BFS, Justus Liebig University, 35392 Giessen, Germany; max.moescheid@vetmed.uni-giessen.de (M.F.M.); simone.haeberlein@vetmed.uni-giessen.de (S.H.); christoph.grevelding@vetmed.uni-giessen.de (C.G.G.); 8Institute of Molecular Pathobiochemistry, Experimental Gene Therapy and Clinical Chemistry (IFMPEGKC), RWTH University Hospital Aachen, 52074 Aachen, Germany; rweiskirchen@ukaachen.de; 9Endemic Medicine Department, Faculty of Medicine, Helwan University, Cairo 11795, Egypt; m_elkassas@hq.helwan.edu.eg; 10Pathology Department, Faculty of Medicine, Mansoura University, Mansoura 35516, Egypt; kzalata@mans.edu.eg

**Keywords:** hepatic stellate cells, parasite, host, fibrosis, metabolism

## Abstract

Schistosomiasis affects over 250 million people worldwide, with the highest prevalence at the age of 10–14 years. The influence of the host’s age on the severity of liver damage is unclear. We infected male 8, 14, and 20-week-old mice with *S. mansoni*. Hepatic damage, inflammation, fibrosis, and metabolism were analyzed by RT-qPCR, Western blotting, ELISA, immunohistochemistry, and mechanistic transwell chamber experiments using *S. mansoni* eggs and human hepatic stellate cells (HSCs) or primary mouse hepatocytes. Major results were validated in human biopsies. We found that hepatosplenomegaly, granuloma size, egg load, inflammation, fibrosis, and glycogen stores all improved with the increasing age of the host. However, serum alanine transaminase (ALT) levels were lowest in young mice infected with *S. mansoni*. Hepatic carbohydrate exploitation was characterized by a shift towards Warburg-like glycolysis in *S. mansoni*-infected animals. Notably, *S. mansoni* eggs stimulated hepatic stellate cells to an alternatively activated phenotype (GFAP^+^/desmin^+^/αSMA^−^) that secretes IL-6 and MCP-1. The reduction of fibrosis in older age likely depends on the fine-tuning of regulatory and inflammatory cytokines, alternative HSC activation, and the age-dependent preservation of hepatic energy stores. The current results emphasize the significance of investigations on the clinical relevance of host age-dependent liver damage in patients with schistosomiasis.

## 1. Introduction

Schistosomiasis is a neglected tropical disease caused by parasitic trematodes of the genus *Schistosoma*. These parasites can cause acute and chronic diseases in the gastrointestinal or urinary tracts of humans and animals. Globally, more than 250 million people in 78 countries require preventive treatment; 90% of these live in Africa [1]. With climate change and globalization, endemic areas are expanding, and schistosomiasis has been found to also spread in regions with moderate climates like Corsica (France) [2] and Almeria (Spain) [3]. Cercariae, a larval form of the parasite, are released by the infected secondary host, a freshwater snail. Cercariae can penetrate the skin of humans in contact with contaminated freshwater. Adult flukes develop, mature, and pair in the host’s circulation, where they produce eggs. Transmission occurs when infected individuals contaminate freshwater with feces or urine containing parasite eggs. These eggs hatch in the water and subsequently infect the secondary host [1]. Acute and chronic tissue damage can be caused by an immune reaction against the parasites’ eggs but also due to direct effects triggered by the eggs in host parenchyma [4,5]. The species *S. mansoni* causes most cases of intestinal and hepatic schistosomiasis, often leading to serious consequences like periportal fibrosis and granuloma formation, hepatosplenomegaly, and portal hypertension [6]. The prevalence of *S. mansoni* infection peaks in children aged 10–14 years, whereas a peak shift towards adolescents and young adults has been reported in low transmission areas with lower overall prevalence [7,8]. Health consequences and potential effects of schistosomiasis in young children are increasingly recognized [9]. The age-prevalence curve is primarily influenced by exposure to contaminated water and individual immunological characteristics of the human host [10]. Despite this, the egg output of *S. mansoni* remains stable regardless of the host’s age or the intensity of the infection [11].

Experimental models of *S. mansoni* infections in mammals have contributed to our understanding of the pathogenesis of infection, particularly in the hepatic formation of fibrotic granulomas [12]. It has been shown that very young, prepubertal mice are more susceptible to infection and yield higher recovery of adult trematodes compared to elder mice [13]. Additionally, cercariae mortality during skin passage steadily increases with the host’s age, and cercariae mortality during skin passage reaches a steady level in adult mice at approximately 28–35 days [13]. The percentage of recovery of adult flukes from infected mice older than 4 weeks was consistent across all ages up to 7 months of the host’s age [14]. It is now widely accepted that early childhood infections play a crucial role in modulating the host’s immunity and the development of chronic anti-schistosomal inflammation. This, in turn, contributes to the major pathologic effects like anemia, hepatosplenomegaly, and hepatic fibrosis observed in pediatric populations in endemic areas [15,16]. In our study, we investigated the age dependency of fibrogranulomatous schistosomiasis in mice and defined its molecular principles.

## 2. Materials and Methods

### 2.1. Human Material

Pseudonymized human liver samples were kindly provided by the Pathology Department, Faculty of Medicine, Mansoura University, Egypt. The use of pseudonymized human residual samples that were routinely taken for pathologic assessment was approved by the local ethics committee (AZ 05/19). According to the ethics vote, informed consent was not required for our retrospective analyses of archived tissues. All experiments were performed in accordance with relevant guidelines and regulations.

### 2.2. Animal Model

We used Biomphalaria glabrata snails as intermediate hosts to maintain the *S. mansoni* life cycle and male black six mice (C57BL/6JCrl, obtained from Charles River) as final hosts. A total of 54 mice were randomly divided into 6 groups. Twelve male mice from each age group (eight, fourteen, and twenty weeks) were infected with 100 cercariae (both sexes) using the pre-soaking procedure in a water bath at 30 °C [17]. Six non infected mice from each age group were treated in the same way without cercariae and used as super controls. After nine weeks of infection, the mice were anesthetized with Isoflurane (4% *v*/*v*, ecuphar GmbH, Greifswald, Germany), subsequently euthanized by cervical dislocation, and the organs were preserved for individual analysis. All animal experiments were conducted in accordance with the European Convention for the Protection of Vertebrate Animals used for experimental and other scientific purposes (ETS No 123; revised Appendix A) and were approved by the Regional Council Giessen (V 54-19 c 20 15 h 01 GI 20/10 Nr. G 44/2019). The study is reported in accordance with ARRIVE guidelines [18].

### 2.3. Histologic Staining and Immunohistochemistry

Liver sections of 3 µm thickness were produced on a microtome from the formalin-fixed and paraffin-embedded caudate lobe. Sirius red staining was performed as described in [19]. Immunohistochemistry (IHC) was performed on liver sections of representative mice using the ImmPRESS AP REAGENT KIT (MP-5401) (Vector Laboratories, Inc., Newark, CA, USA). Deparaffination, unmasking, and blocking were performed as described before [20]. The primary antibodies used were rabbit anti desmin IgG (PA5-16705, Life Tech, Darmstadt, Germany). The color reaction was developed using the Permanent AP Red Kit (ZYT-ZUC001-125) from Zytomed Systems (Berlin, Germany).

### 2.4. Assessment of Granuloma Extent and Fibrotic Areas

High-resolution scans of liver slices from each mouse were assessed using a NanoZoomer (Hamamatsu Photonics, Fukuoka City, Japan). The granuloma area or Sirius red stained area was color labeled and quantified in ImageJ, version 1.51 [21].

### 2.5. Quantitative Real-Time PCR

mRNA isolation was performed using an RNeasy^®^ Mini Kit from Qiagen (#74106, Qiagen N.V., Venlo, The Netherlands) according to the manufacturer’s protocol. cDNA was transcribed using the iScript cDNA Synthesis Kit (#1708891, Bio-Rad Laboratories, Inc., Hercules, CA, USA) according to the manufacturer’s protocol. Primer sequences are located in the Appendix A.

### 2.6. Serum Analysis, Hydroxyproline Assay, and Potassium-Hydroxide Digestion

The ALT serum parameter was analyzed using a DiaSys respons^®^910 Chemistry Analyzer, following the manufacturer’s protocol (#1 2701 99 10 920, Diasys, Holzheim, Germany). The assessment of hydroxyproline was conducted as described before [22]. The liver tissue was digested for 16 h in a 5% potassium hydroxide solution (KOH) at 37 °C, and the remaining eggs were counted under a microscope as described before [23].

### 2.7. Cell Culture

LX-2 cells were kindly provided by Scott L. Friedman, New York, USA [24]. After 24 h of seeding, 200 *S. mansoni* eggs were added to the wells in a cell culture insert (Corning Costar, Glendale, AZ, USA, Transwell #3421). LX-2 cells were stimulated in co-culture with *S. mansoni* eggs for 24 h. The supernatants of the individual wells were analyzed via ELISA. The LX-2 cells were lysed for Western blot analysis.

### 2.8. Enzyme-Linked Immunosorbent Assay (ELISA)

To quantify CCL2 and IL-6 concentrations in cell culture supernatants from *S. mansoni* egg-stimulated HSCs, an Enzyme-linked Immunosorbent Assay (ELISA) was performed in technical duplicates using ELISA kits from R&D Systems, Minneapolis, MN, USA. The assay ranges for the ELISA kits were 15.6–1000 pg/mL (for MCP-1/CCL2) and 9.4–600 pg/mL (for IL-6).

### 2.9. Statistical Analysis

Statistical analysis of the current exploratory study was performed with SPSS 29.0. (SPSS Inc., IBM Corp., Armonk, NY, USA). Dependencies between metric parameters were calculated by curve fitting and the selection of best-fitting models. A one-way ANOVA and post hoc Fisher’s LSD test were used for pairwise comparison between groups. Relevant *p*-values are indicated in the figures, while the probabilities of differences between infected mice and the corresponding control were denoted as * for *p* < 0.05 and ** for *p* < 0.001, as indicated in the legend.

## 3. Results

### 3.1. An Older Age at the Time of Infection Reduced Hepatic Egg Load, Granuloma Extent, and Liver-to-Body Weight Ratio but Increased Serum ALT

The successful infection of mice of different ages (Figure 1A: eight weeks, *n* = 7 out of 12; fourteen weeks, *n* = 10 out of 12; twenty weeks, *n* = 8 out of 12) was validated nine weeks after infection by microscopic detection of *S. mansoni* eggs in stool and liver tissues. The effectiveness of infection is depicted in Appendix A. Mice incubated with cercariae without successful infection and one mouse with eggs only in one liver lobe were excluded from further analysis.

The infection manifested itself in reactive inflammatory granulomatous liver changes characterized by the uneven hepatic surface (Appendix A) and hepatomegaly. Three of the infected mice died. However, the autopsy gave no evidence of schistosomal involvement in their death or an unambiguous cause of death. The total extent of granulomas and hepatic egg load was reduced in both animal groups infected at older ages (14 weeks, 20 weeks) compared to the animals infected at eight weeks (Figure 1B,C and Appendix A). There were no differences between the two groups of mice infected at an older age in terms of the spread of granulomas and hepatic egg burden. Liver-to-body weight ratios increased after infection regardless of age, but they were less pronounced when the hosts were infected at an older age (Figure 1D). The hepatic egg load correlated positively with the liver-to-body weight ratio (Figure 1E). Serum ALT levels were elevated at all ages after the infection (Figure 1F). In contrast to granuloma size, egg load, and liver-to-body weight ratio, the extent of the ALT increase was relatively low when younger animals (8 weeks old) were infected (Figure 1F). Due to this, the inverse dependency of serum ALT and the liver-to-body weight ratio appears at least unusual (Appendix A). Accordingly, the correlation between hepatic egg load and serum ALT even depicted a moderate inverse trend (Figure 1G). Of note, this inverse trend was reproduced across all test groups, i.e., all age groups, as indicated by the color code (Figure 1G).

### 3.2. S. mansoni-Induced Splenomegaly Was More Pronounced in Young Animals

*S. mansoni* eggs were found in the spleens of 36% of the mice that were successfully infected (Figure 2A). However, the presence of splenic eggs did not affect serum ALT levels. It is worth noting that the liver-to-body weight ratio was higher in mice with eggs in the spleen (Figure 2B). Notably, spleen weight (Figure 2C) and spleen-to-body weight (Figure 2D) decreased with infections at an older age.

Lymphatic hyperplasia (Figure 2E) and extramedullar hematopoiesis (Appendix A) increased in the spleen of older mice infected with *S. mansoni*. Splenic granulomas, mostly appearing with a circular shape in histology (Appendix A), were irregularly interspersed with fibrillary collagen fibers (Figure 2F).

### 3.3. S. mansoni Induced Granulomatous Hepatic Inflammation Impaired Younger Animals with More Intensity

Leukocyte infiltration and the transcriptional regulation of inflammatory cytokines were assessed to identify the involvement of inflammatory processes in hepatic fibrogranulomatous alterations. After nine weeks of infection with *S. mansoni*, inflammatory cells accumulated around the schistosome eggs, forming hepatic granulomas (Figure 3A, dashed line). In addition to these granulomatous aggregates, smaller inflammatory infiltrates appeared within the parenchyma (Figure 3A, arrowhead) and in close proximity to flukes in large branches of the portal vein (Appendix A). The infection led to an increase in hepatic inflammation, as demonstrated by the analysis of *Cd45* (Figure 3B). Hepatic levels of *Cd45* remained comparable in all age groups of uninfected animals, as well as in infected animals, respectively, but were consistently elevated in infected mice compared with healthy mice of the same age groups (Figure 3B).

Hepatic expression of the proinflammatory cytokines *Tnfα*, *Ifnγ*, and *Il12* increased with infection and decreased with the host’s age at the time of infection (Figure 3C,D and Appendix A). Moreover, the Th2-specific cytokine *Il4* and the immunomodulatory cytokine *Il10* were induced with infection and most pronounced when the infection occurred at an early age (Figure 3E,F). Notably, among all analyzed cytokines, *Il10* expression showed the highest dependency with the hepatic egg load (Appendix A). In contrast to the other cytokines analyzed, the expression of the autoimmune-associated and TH17-activating cytokine *Il23* appeared to be independent of age (*p* = 0.067, Appendix A).

### 3.4. S. mansoni-Induced Hepatic Fibrosis Is Most Pronounced in Young Infected Animals

Host age-related effects of hepatic fibrogenesis were analyzed to address its role in hepatic long-term damage during schistosomiasis. Hepatic hydroxyproline levels were increased by *S. mansoni* infection and decreased with an older host’s age at the time of infection (Figure 4A). Furthermore, there was a strong dependency between hepatic egg load and hepatic hydroxyproline level (Figure 4B).

The two most abundant fibrillar collagens in liver fibrosis, type I and type III collagen, were induced at the transcriptional level by *S. mansoni* infection, with the highest levels observed in mice infected at 14 weeks of age (Figure 4C and Appendix A). Sirius red staining was used to visualize granulomatous hepatic fibrosis (Figure 4D). Morphometric quantification of the Sirius red-stained areas also showed an accumulation of fibrillary collagen in the livers of *S. mansoni*-infected mice (Appendix A). Consistent with the hydroxyproline levels, the hepatic gene expression of the profibrotic cytokines *PdgfB* and *Tgfβ* was induced by *S. mansoni* infection and dropped with increasing age of the infected host (Appendix A). The protein level of αSMA, a marker for the transdifferentiation of hepatic stellate cells to myofibroblasts, was not regulated (Figure 4E). However, the level of desmin, a marker for activated and transdifferentiated HSCs [25], increased after infection in all age groups (Figure 4F). Immunostaining of mouse liver slices confirmed the increasing number of desmin-positive cells in the liver parenchyma and within the granuloma (Appendix A). In order to obtain a model-independent proof of the results from mouse and human cell lines, desmin^+^ cells were visualized in granuloma of a histologic slice of a liver biopsy that was routinely taken for diagnosis from a 31-year-old male patient suffering from schistosomiasis (Appendix A).

### 3.5. S. mansoni Egg Activated HSCs towards a GFAP^+^/desmin^+^/αSMA^−^ Phenotype Expressing IL-6 and MCP-1

As the infection with *S. mansoni* induced hepatic desmin but not αSMA, we aimed to analyze whether *S. mansoni* eggs directly stimulate alternative HSC activation without an immune reaction against the parasite. Therefore, human LX-2 cells [24] were cocultivated with increasing numbers of *S. mansoni* eggs in a transwell chamber system (Figure 5A). Of note, expression levels of the transdifferentiation marker αSMA decreased when co-cultured with *S. mansoni* eggs, while GFAP and desmin were induced in LX-2 (Figure 5A–D). Furthermore, cocultivation with increasing numbers of *S. mansoni* eggs led to a more pronounced activation of GFAP and desmin expression (Figure 5C,D). αSMA and desmin are reliable markers of activated and myofibroblast-like HSCs [25,26]. It has been reported that exposure of LX-2 cells to IL-1, TNF-α, and IL-8 can reverse the phenotype of pro-fibrogenic activated cells [27]. The unusual αSMA-/desmin^+^ phenotype of *S. mansoni*-activated HSCs raised the question of whether these HSCs were also characterized by an altered cytokine release as described for LX-2 cells [27]. Therefore, we analyzed the expression of IL-6 and MIP-1/CCL2 by ELISA. Indeed, protein expression and release of both factors were induced in *S. mansoni*-activated HSCs (Figure 5E,F).

### 3.6. S. mansoni Egg-Exhausted Glycogen Stores Improved with Increasing Host Age

Recently, we demonstrated that *S. mansoni* eggs have the ability to reprogram hepatic lipid and carbohydrate metabolism through soluble factors, ultimately leading to oxidative stress in the host’s parenchyma [4]. Hepatic glycogen stores were also exhausted in *S. mansoni*-infected mice and particularly strong in younger animals (Figure 6A). Hepatic glycogen content is inversely correlated with the number of eggs in the liver tissue (Figure 6B). The induction of key enzymes in glycolysis was another characteristic of dysregulated hepatic carbohydrate metabolism observed previously [4]. In the current study, *S. mansoni* infection caused a Warburg-like glycolysis [28,29] characterized by an induction of *Hk2*, *Pkm2*, and *G6pdh* (Figure 6C–E). Notably, the induction of the pentose phosphate pathway, as indicated by *G6pdh* levels, correlated well with the hepatic expression of *Il4* (Figure 6F). As the infection with *S. mansoni* induced hepatic *Hk2*, *Pkm2*, and *G6pdh*, we aimed to analyze whether *S. mansoni* eggs directly stimulate these enzymes without an immune reaction against the parasite. For this purpose, primary mouse hepatocytes (PMH) [30] were cocultivated with increasing numbers of *S. mansoni* eggs in a transwell chamber system. *Hk2* and *Pkm2* increased, while *G6pdh* was reduced in *S. mansoni* egg-stimulated PMH (Appendix A).

Furthermore, *S. mansoni* eggs were shown to be capable of mobilizing and exploiting hepatic lipid stores [4]. Perilipin 2 (*Plin2*) promotes the formation of lipid droplets (LDs), and hepatic PLIN2 expression levels correlate with triacylglyceride (TAG) content [31]. In this study, we demonstrated that hepatic expression of *Plin2* decreased in *S. mansoni*-infected mice, regardless of the host’s age (Appendix A). Moreover, the expression of *Fasn*, a key enzyme involved in de novo lipid synthesis, was reduced in the liver of *S. mansoni*-infected mice (Appendix A). However, no age-related differences were observed in fat metabolism changes.

Finally, the hepatic gene expression of catalase (*Cat*), an enzyme protecting against oxidative damage, was diminished in *S. mansoni*-infected mice (Appendix A). The host’s age had no effect on *Cat* regulation. Hepatic *Cat* levels positively correlated with serum ALT in *S. mansoni*-infected mice (Appendix A). In this regard, the counter-regulation of hepatic damage is also maintained in infected mice of all ages.

## 4. Discussion

Schistosomiasis is a poverty-related and neglected tropical disease (PRNTD) with the largest burden of disease in sub-Saharan Africa [32]. In endemic areas with high incidence, schistosomiasis commonly affects the poor, who have limited access to clean water, adequate sanitation, and hygiene services [33]. In Brazil, a higher prevalence of schistosomiasis was found in males aged 10–39 years, likely due to leisure or work activities that involve contact with contaminated water [34]. Recent studies, including meta-analyses from Ethiopia, revealed a higher prevalence of *S. mansoni* infection among male children of school-age [35,36,37,38]. Regionally, even a very high prevalence of *S. mansoni* infections among preschool children has been observed in certain regions [39,40]. The local accumulation of schistosome transmission suggests that it depends on socio-demographic, ecological, behavioral, and household factors [38,41]. However, factors determining susceptibility to infection, such as sex [14], immunology [42], the host’s microbiome [43], genetics [44], and an altered barrier function of aged skin [13] may play additional causative roles. It has been demonstrated that the number of cercariae dying in the skin of very young mice during penetration (2 days old) is less than one-third as high as in adult mice [13]. Parasite losses in the skin increase with the age of the host and reach a steady level in adult mice at about 28–35 days [13]. Additionally, the burden of adult flukes in mice older than 4 weeks was found to be similar at all ages up to 7 months [26]. Based on these considerations and the fact that male animals have a significantly higher parasite burden than females [14], we aimed to identify the impact of the host’s age on the molecular and cell biology of schistosomal hepatosplenic granuloma formation in male mice infected at different age levels.

The hepatic egg burden and extent of granuloma were found to be higher in younger hosts. This increase, indicated by an inverse dependency on age, was associated with a higher liver-to-body weight ratio (Figure 1B–E). Previous studies showed that *S. mansoni* egg excretion remains stable regardless of the age of the host or the intensity of infection [11]. In addition, the higher liver weight appeared not only to depend on the egg load but also on the size of the single granuloma. This apparent discrepancy could be attributed to differences in egg production between older mice and humans, which has not been analyzed previously.

Beyond that, it is unknown if the hepatic egg load, in contrast to excretion, might be influenced by the host’s age. Only 36% of mice with infestation of the whole liver showed eggs in the spleen (Figure 2A). However, the reduction in spleen weight with age parallels our findings from the liver. Spleen weight and spleen weight per body weight are highest in younger animals. Of note, the increase in serum ALT (Figure 1F and Figure 2B) depends less or not at all on the spleen infestation but is inversely correlated to the egg load (Figure 1G). It has been widely accepted that liver damage is provoked by the eggs itself and by the fibrogranulomatous immune reaction of the host [4,45]. The increase in serum ALT was relatively low, and the absolute levels in *S. mansoni*-infected mice were around ten times lower than in mouse models for chronic toxically-induced or cholestatic liver diseases [46]. Therefore, we believe that hepatocellular damage, as reflected by serum ALT, in the early stages of chronic hepatic schistosomiasis (nine weeks after infection in mice) may not be directly correlated with egg load and the extent of granuloma formation (Figure 1G). Increasing ALT over time may, for example, depend on increased lymphatic hyperplasia observed mainly in older animals (Figure 2E).

CD45 is expressed on all nucleated hematopoietic cells and plays a central role in antigen receptor signal transduction and lymphocyte development [47]. Hepatic gene expression of the pan-leukocyte marker *Cd45* increased after infection, but it was not influenced by the host’s age at the time of infection (Figure 3B). As the hepatic level of *Cd45* may be used as a measure of inflammatory infiltration of the liver, it appears likely that, apart from granulomas, smaller accumulations of leukocytes within the parenchyma contribute to hepatocellular damage (Figure 3A and Appendix A). Notably, the hepatic expression of type I and type II cytokines like Tnfα, *Ifnγ*, *Il12*, or *Il4,* as well as regulatory cytokines like *Il10,* showed maximum expression at the youngest age of infection similar to egg load and granuloma extent. IL-10 was designated as the central immunomodulatory regulator in the pathogenesis of schistosomiasis [48]. Both type 1 and type 2 cytokines were excessively triggered by an IL-10 knockout in *S. mansoni*-infected mice, which led to enhanced fibrogranulomatous damage and increased mortality [48]. Moreover, IL-10/IL-4- and IL-10/IL-12-deficient mice developed highly polarized type 1 and type 2 cytokine responses, respectively. The *S. mansoni*-infected double knockout mice either experienced intense inflammation (IL-10/IL-4-type I polarized) or extensive fibrosis (IL-10/IL-12-type II polarized), both leading to increased mortality [48]. The expression levels of central regulators *Il4*, *Il10*, and *Il12* in the liver showed their maximum expression in young infected mice (Figure 3E,F and Appendix A). Considering the effects observed in IL-10/IL-12-deficient mice, the improvement of fibrotic granulomas (Figure 1B–D) seemed to be dependent on the fine-tuning of *Il4*, *Il10*, and *Il12*. Moreover, we hypothesize that the reduction of fibrosis with increasing host age (Figure 5) and the regulation of *Il4*, *Il10*, and *Il12* may be mutually dependent. In contrast to these findings, lymphatic hyperplasia in spleens (Figure 2E) was increased in older animals. Although both effects, the reduced expression of essential immune-modulating cytokines in the liver and the rise of lymphatic hyperplasia in the spleen, occur simultaneously in older infected animals, it is unclear if they depend on each other. It is also possible that these results reflect an age-dependent compensatory immune reaction against the parasites.

It has been previously demonstrated that *S. mansoni* infection blocks bone marrow erythropoiesis and increases spleen erythropoiesis [49], which is consistent with the anemia and splenomegaly commonly seen in schistosomiasis patients [50]. This is likely a compensatory splenic response to the depletion of bone marrow hematopoiesis caused by the feeding of schistosome flukes [51]. We also observed increased extramedullary hematopoiesis in the spleens of older animals (Appendix A). Therefore, we propose that the host’s age may contribute to the impairment of the medullar hematopoietic system to regenerate upon *S. mansoni* infection, as shown before [49].

It has been suggested that schistosome eggs have a significant influence on HSCs in their immediate vicinity, inhibiting fibrogenesis and causing HSCs to adopt a more quiescent phenotype [52]. Another study demonstrated that the *S. japonicum* T2 RNase protein significantly reduced the expression levels of αSMA and Smad4 in the human hepatic stellate cell line LX-2 [53]. Increased synthesis of desmin and the formation of desmin-containing intermediate filaments are characteristic of the transdifferentiation of HSCs into myofibroblast-like cells [25]. Our data suggest that the observed αSMA^−^/desmin^+^ phenotype in *S. mansoni*-infected mice and egg-stimulated LX-2 cells represent no quiescent HSC phenotype but rather an alternatively activated HSC phenotype [25] localized in the granuloma at the site of fibrogenesis (Appendix A). Importantly, egg-stimulated αSMA^−^/desmin^+^/GFAP^+^ LX-2 cells produced even higher levels of IL-6 and MCP-1/CCL2 (Figure 5E,F), comparable to the change in phenotype of pro-fibrogenic activated LX-2 cells exposed to TNFα [27]. GFAP is associated with the acquisition of contractile properties in liver stellate cells, which are more closely associated with the early stages of fibrosis [54]. Since GFAP expression depends on the egg load, and the egg load is higher in young infected animals, the latter are more affected by fibrosis.

In a clinical study, young adults between the ages of 23 and 33 had the highest prevalence of *S. mansoni*-induced hepatic fibrosis in a cohort of 199 individuals aged 6 to 50 residing in a Ugandan village [10]. Furthermore, the risk of fibrosis was associated with different cytokine profiles depending on age and gender. Generally, a high risk of fibrosis was linked to high TNF-α factor scores in adults [10]. A population-wide study in northern Senegal showed that the severity of *S. mansoni*-specific fibrosis peaks in adults aged 20–29 and decreases in older patients [55]. In the current study, the youngest infected mice had the highest levels of hepatic hydroxyproline and *Tnfα* expression, which both decreased in mice infected at an older age. Our observations from the mouse model align with the clinical findings, as the physiology of mice at 8 weeks is comparable to that of young human adults [56]. Therefore, the model and data presented here provide a relevant foundation for further investigations, as liver fibrosis and more severe outcomes from chronic hepatic injury are crucial for disease prognosis and patient stratification.

It is well-known that liver fibrosis has a profound impact on T cell activation. Consequently, the different inflammatory reactions could be a secondary effect of fibrosis, which reduces the ability of immune cells to perform their effector functions. For instance, liver fibrosis is characterized by a drastic reduction in endothelial porosity, which is one of the major reasons for reduced intrahepatic immune reactions [57,58].

Finally, the lower glycogen stores of the younger animals (Figure 6A) fit well with the more extensive fibrogenesis and granuloma formation in the case of schistosomiasis. Of note, the induction of hepatic Warburg-like glycolysis (Figure 6C–E) [28,29] in *S. mansoni*-infected animals appears to be a novel aspect of the parasite-host-parenchyma interaction, which may help in understanding the metabolic reprogramming of the host [4]. Similar to hepatitis C [28], *S. mansoni* eggs are also capable of inducing the so-called aerobic glycolysis or Warburg effect, which serves to provide more anabolic metabolites upstream of the citric acid cycle, such as amino acids, pentoses, and NADPH, for cell growth and egg survival. It should be noted that the induction of *Hk2* and *Pkm2* expression in hepatocytes was also dependent on the hepatic egg load (Appendix A) and, therefore, may have a particularly strong effect on young animals. As G6PDH was mainly expressed by granuloma cells in *S. mansoni*-infected hamsters [4], the currently observed hepatic expression of G6pdh seems to reflect the expansion of granulomatous tissue in infected mice (Figure 6E). It still remains speculative why the pentose phosphate pathway was reduced in primary mouse hepatocytes co-cultured with eggs from *S. mansoni* (Appendix A).

Limitations: We refrained from recovering adult parasites by perfusion as a stable burden has been described in adult mice [14]. Rather, we used the serum to analyze the ALT level as a measure of hepatocellular damage. The availability of human samples for validation is limited, and the presence of desmin^+^ cells in granulomas of human patients should be confirmed in future studies with additional material.

In conclusion, our results provide compelling evidence that the age at the time point of infection determines the severity of the pathologic consequences.

## Figures and Tables

**Figure 1 cells-13-01643-f001:**
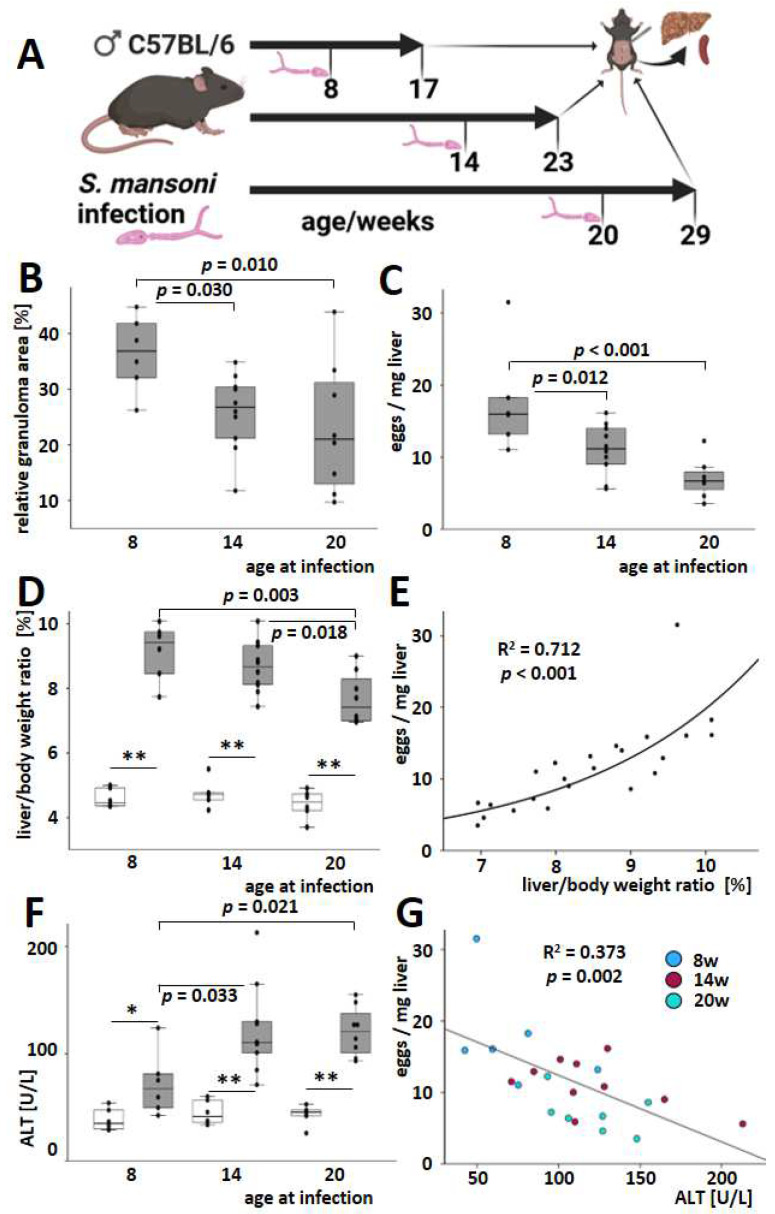
Hepatic damage and egg load in *S. mansoni*-infected mice. (**A**) Schematic illustration of the animal experiment. Created in BioRender. (**B**) The expansion of granulomatous tissue decreased in older hosts. Granuloma areas were assessed, as shown in Appendix A. (**C**) Hepatic egg load and liver-to-body weight ratio (**D**) were reduced in older *S. mansoni*-infected mice. (**E**) The liver-to-body weight ratio correlated well with hepatic egg load in *S. mansoni*-infected animals. Curve fitting was performed to assess the coefficient of determination and the related probability. (**F**) Serum ALT levels were raised by *S. mansoni* infection in all age groups. When animals were infected with *S. mansoni* at the age of 14 weeks, ALT values increased more prominently compared to the younger animals. There were no changes compared to the older animals. (**G**) Beyond that, serum ALT was inversely correlated with hepatic egg load across all age groups of *S. mansoni*-infected mice. Curve fitting analysis was performed to calculate statistical parameters. Color codes indicate the age of infection: blue for 8 weeks, red for 14 weeks, and green for 20 weeks. * indicates *p* < 0.05, ** indicates *p* < 0.001. White bars represent uninfected mice, while grey bars represent infected mice. Uninfected controls: *n* = 6, *S. mansoni* infected animals: *n* = 6 (8 w), 10 (14 w), and 8 (20 w). The indicated *p*-values were calculated by curve fitting analysis or ANOVA and post hoc pairwise comparison of groups using Fisher’s LSD.

**Figure 2 cells-13-01643-f002:**
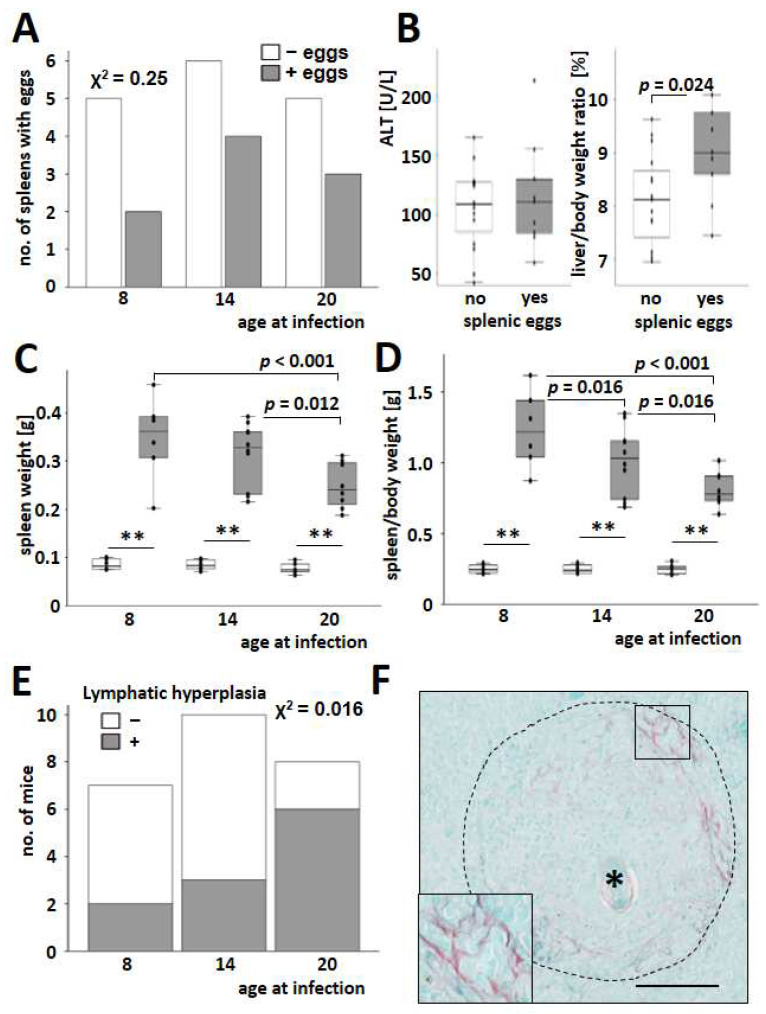
*S. mansoni*-induced splenomegaly in infected mice. (**A**) Rate of histologically positive findings for eggs in the spleen (grey bars) of infected mice. White bars indicate a negative histological result for eggs in the spleen. Chi-square tests over all groups showed no differences. (**B**) Infected mice with eggs in the spleen (grey bars) had a higher liver-to-body weight ratio than mice without splenic eggs (white bars). (**C**,**D**) Spleen weight (**C**) and spleen-to-body weight ratio (**D**) were increased in infected mice (grey bars) and reduced in older infected animals. (**E**) Lymphatic hyperplasia was assessed histologically by an experienced veterinary pathologist (KK). The number and size of lymphoid follicles, as well as the cellularity of red pulp and occurrence of extramedullary hematopoiesis, were assessed semiquantitatively. Inflammatory lesions were characterized, and parasitic stages were counted. (**F**) Sirius red staining demonstrated splenic fibrosis within the granuloma around *S. mansoni* eggs (*). Magnification 200×, bar 100 µm. Uninfected controls: *n* = 6, *S. mansoni* infected animals: *n* = 6 (8w), 10 (14w), and 8 (20w). ** indicates *p* < 0.001; white bars represent uninfected mice. The indicated *p*-values were calculated using the Chi^2^-test or ANOVA and a post hoc pairwise comparison of groups using Fisher’s LSD.

**Figure 3 cells-13-01643-f003:**
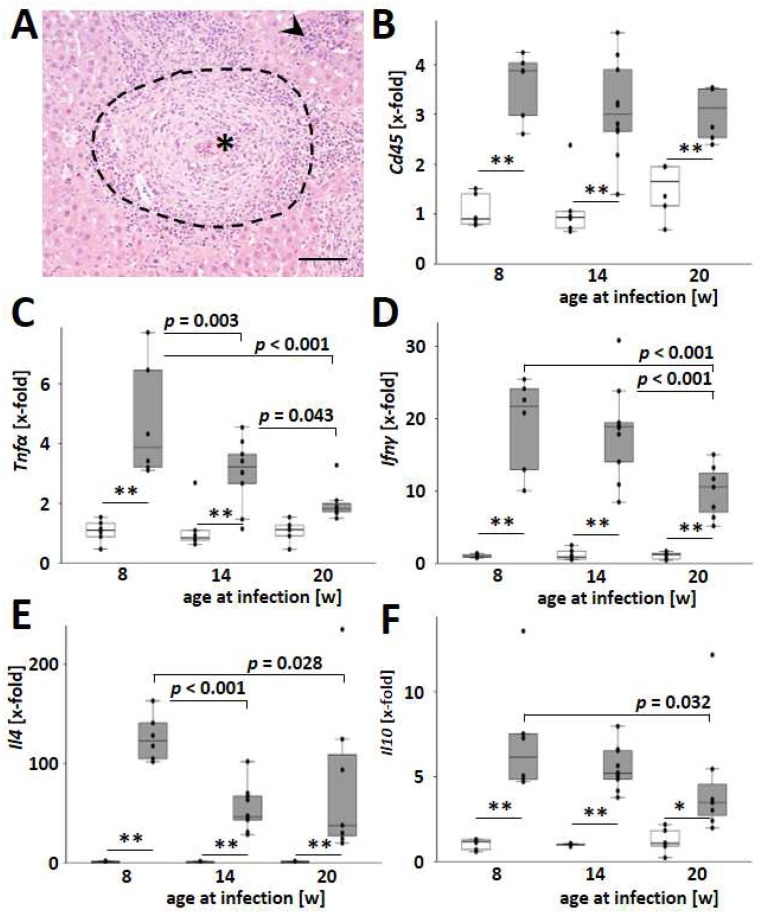
*S. mansoni*-induced hepatic cytokine expression was highest in younger animals. (**A**) Inflammatory infiltration led to the formation of granulomas (indicated by dashed line) around *S. mansoni* eggs (*) and the accumulation of leukocytes within the liver tissue (indicated by arrowhead). A representative liver slice stained with H&E is shown, magnification 200×, scale bar 100 µm. (**B**) Hepatic mRNA levels of the leukocyte marker *Cd45*. (**C**–**F**) Hepatic mRNA levels of the cytokines *Tnfα*, *Ifnγ*, *Il4*, and *Il10* increased with infection but decreased with the age of the host at the time of infection. Uninfected controls: *n* = 6, *S. mansoni* infected animals: *n* = 6 (8w), 10 (14w), and 7 (20w, one sample from this group was excluded from qPCR due to mRNA degradation) and 3 technical replicates each. ** indicates *p* < 0.001, white bars represent uninfected mice, and grey bars are infected mice. The indicated *p*-values were calculated by ANOVA and post hoc pairwise comparison of groups using Fisher’s LSD.

**Figure 4 cells-13-01643-f004:**
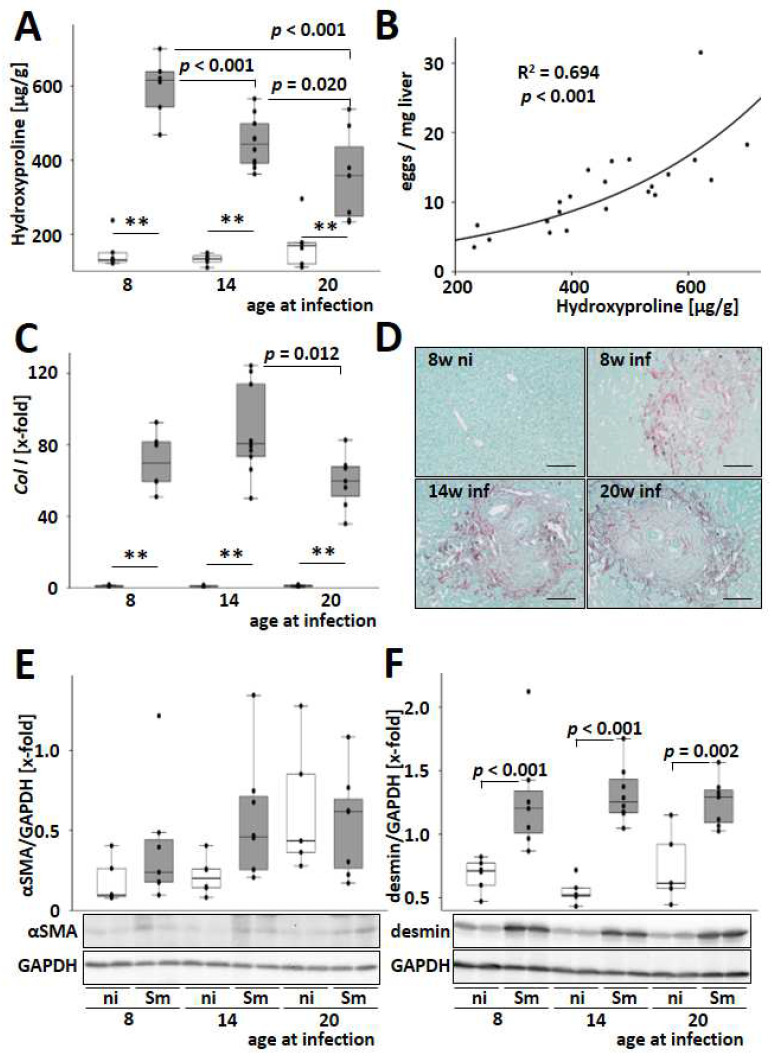
*S. mansoni*–induced hepatic fibrosis is highest among the younger infected. (**A**) *S. mansoni* infection-induced hepatic hydroxyproline content, which is a quantitative measure of fibrillar collagen accumulation and, therefore, hepatic fibrosis. Due to the material-consuming analysis and its good reproducibility, hydroxyproline quantification was performed once only. (**B**) The number of eggs/mg liver tissue correlated well with hepatic hydroxyproline level in *S. mansoni*-infected animals as determined by curve fitting analysis. (**C**) Hepatic mRNA levels of type I collagen, *Col I*, were induced by *S. mansoni* infection. (**D**) Representative liver slices stained with Sirius red/fast green visualize granulomatous fibrosis. Magnification 200×, scale bar 100 µm. (**E**,**F**) The hepatic protein content of αSMA and desmin were analyzed by Western blotting and subsequent densitometric assessment of signal intensities. Representative blots are shown. Uninfected controls: *n* = 5–6, *S. mansoni* infected animals: *n* = 6 (8w), 8–10 (14w), and 7 (20w) and 3 technical replicates each. ** indicates *p* < 0.001, white bars represent uninfected mice, and grey bars represent infected mice. The indicated *p*-values were calculated by curve fitting analysis or ANOVA and post hoc pairwise comparison of groups using Fisher’s LSD.

**Figure 5 cells-13-01643-f005:**
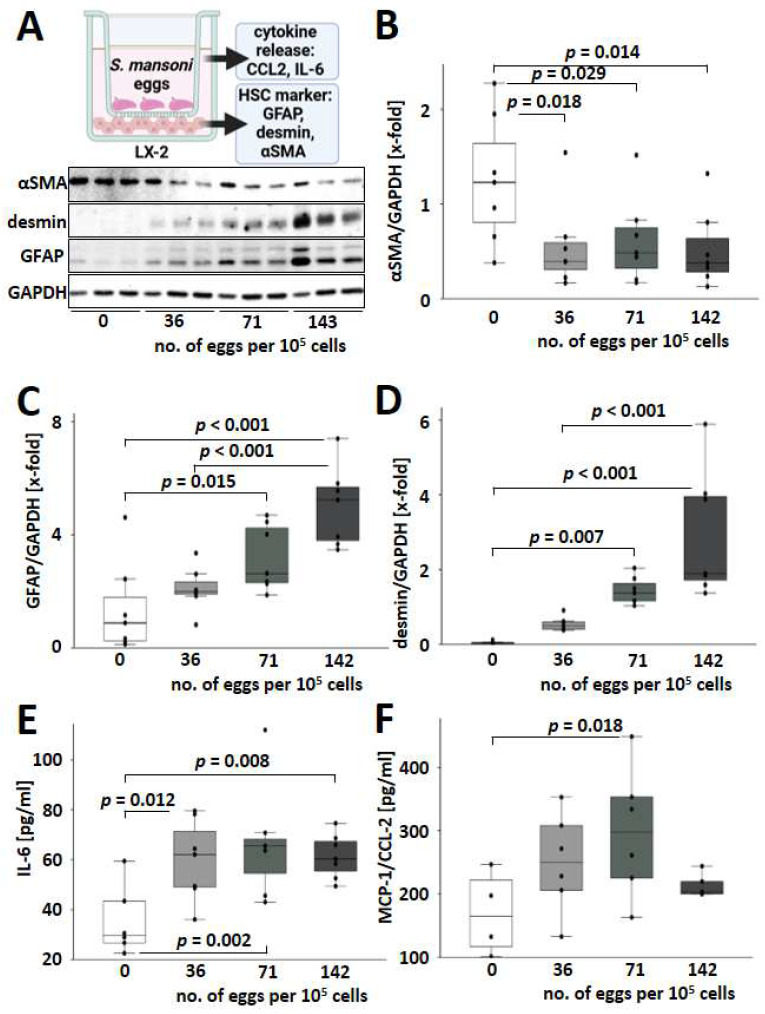
*S. mansoni* eggs activated HSCs towards a GFAP^+^/desmin^+^/αSMA^−^ phenotype. (**A**) Schematic abstract of the transwell co-culture experiments with *S. mansoni* eggs and the human hepatic stellate cell line LX-2 for subsequent analysis of proliferation and transdifferentiation markers, as well as secreted factors. Created in BioRender. Representative Western blots for HSC markers are depicted below. Quantification of HSC markers is presented in box and whisker plots: (**B**–**D**) *S. mansoni* eggs decrease αSMA expression (**B**) in LX-2 cells and induced GFAP (**C**) and desmin (**D**) expression in a dose-dependent manner. (**E**,**F**) Protein levels of human IL-6 and MCP-1/CCL-2 in the supernatant of *S. mansoni* egg-stimulated LX-2 cells. White bars represent no eggs, while grey bars represent increasing numbers of eggs. Collagen I coated 24 well plates, 1.4 × 10^5^ cells per well cultured overnight before 48 h co-culture in DMEM, Transwell: Costar #3421. *n* = 6–7 and 3 technical replicates. The indicated *p*-values were calculated by ANOVA and post hoc pairwise comparison of groups using Fisher’s LSD.

**Figure 6 cells-13-01643-f006:**
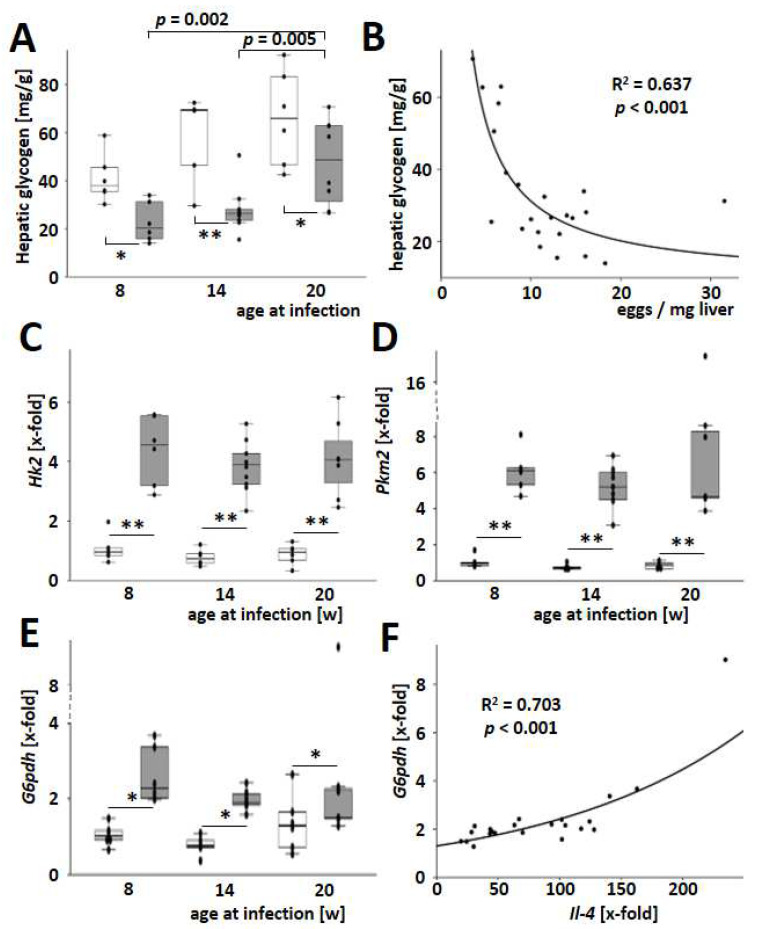
Host age-dependent dysregulation of the hepatic carbohydrate metabolism by *S. mansoni* eggs. (**A**) Total liver glycogen increased with age but was reduced by infection with *S. mansoni*. (**B**) Hepatic glycogen was inversely dependent on the number of *S. mansoni* eggs. (**C**–**E**) Hepatic mRNA levels of hexokinase 2, *Hk2* (**C**), pyruvate kinase M2, *Pkm2* (**D**), and glucose-6-phosphate dehydrogenase, *G6pdh* (**E**) were induced by *S. mansoni* infection. (**F**) Notably, *G6pdh* expression was dependent on *Il4* expression. Curve fitting analysis was performed. Uninfected controls: *n* = 6, *S. mansoni* infected animals: *n* = 6 (8w), 10 (14w), and 7 (20w) and 3 technical replicates each. * indicates *p* < 0.05, ** indicates *p* < 0.001, white bars represent uninfected mice, while grey bars represent infected mice. The indicated *p*-values were calculated using curve fitting analysis or ANOVA and post hoc pairwise comparison of groups using Fisher’s LSD.

## Data Availability

The data that support the findings of this study are available from the corresponding author upon reasonable request.

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
