# Peer review of "Liver Fibrosis Is Enhanced by a Higher Egg Burden in Younger Mice Infected with *S. mansoni"

_cells, 2024, doi:10.3390/cells13191643_

Round 1
Reviewer 1 Report
Comments and Suggestions for Authors
Mueller et al. deliver a very interesting study on a mouse model of S. mansoni. Specifically, they found that in younger, but not older mice, liver fibrosis correlates with egg burden. This notion, of course, needs to be confirmed in human patients, but in principle, it is relevant for disease prognosis and patient stratification.
In this perspective, it would be interesting to include human data from public datasets (if available) or discuss this point based on the current literature.
It is disappointing that the data related to the inflammatory reaction are mainly derived from qPCR performed on total liver lysate. This approach does not allow for discrimination of which cell type is involved in specific gene regulation. For instance, the authors correlate the increased CD45 expression with increased inflammation. This is not entirely accurate. Indeed, CD45 is abundantly expressed by Kupffer cells, so the increased CD45 expression could be related to the recruitment or proliferation of new Kupffer cells, which is not necessarily indicative of inflammation but also occurs during tissue regeneration. Moreover, the increased CD45 expression could result from an increased percentage of Kupffer cells compared to other non-parenchymal cells. Finally, hepatic non-immune cells, such as liver sinusoidal endothelial cells (LSECs), also express CD45.
To correlate the increased CD45 expression with recruitment of inflammatory cells, the authors should perform FACS analysis or PCR analysis on isolated liver immune cells rather than total liver lysates. Alternatively, the authors must rephrase and tone down their comments on this specific aspect.
Finally, it is well-known that liver fibrosis has a profound impact on T cell activation, and consequently, the different inflammatory reactions could be a secondary effect of the fibrosis, which reduces the ability of immune cells to perform their effector functions. For instance, liver fibrosis is characterized by a drastic reduction in endothelial porosity, which is one of the major reasons for reduced intrahepatic immune reactions (PMID: 26188075, 19033667). This aspect should at least be mentioned in the discussion and/or introduction.
Author Response
Response to Reviewer 1
Mueller et al. deliver a very interesting study on a mouse model of S. mansoni. Specifically, they found that in younger, but not older mice, liver fibrosis correlates with egg burden. This notion, of course, needs to be confirmed in human patients, but in principle, it is relevant for disease prognosis and patient stratification.
In this perspective, it would be interesting to include human data from public datasets (if available) or discuss this point based on the current literature.
Authors reply: We would like to thank the reviewer for the kind words about our manuscript and for the helpful suggestions. In response to this suggestion, we have added the following chapter to the discussion section of the revised manuscript to address this point:
´In a clinical study, young adults between the ages of 23 and 33 had the highest prevalence of S. mansoni-induced hepatic fibrosis in a cohort of 199 individuals aged 6 to 50, residing in a Ugandan village [10]. Furthermore, the risk of fibrosis was associated with different cytokine profiles depending on age and gender. Generally, a high risk of fibrosis was linked to high TNF-α factor scores in adults [10]. A population-wide study in northern Senegal showed that the severity of S. mansoni-specific fibrosis peaks in adults aged 20-29 and decreases in older patients [53]. In the current study, the youngest infected mice had the highest levels of hepatic hydroxyproline and Tnfα expression, which both decreased in mice infected at an older age. Our observations from the mouse model align with the clinical findings, as the physiology of mice at 8 weeks is comparable to that of young human adults [54]. Therefore, the model and data presented here provide a relevant foundation for further investigations as liver fibrosis and more severe outcomes from chronic hepatic injury are crucial for disease prognosis and patient stratification.´
It is disappointing that the data related to the inflammatory reaction are mainly derived from qPCR performed on total liver lysate. This approach does not allow for discrimination of which cell type is involved in specific gene regulation. For instance, the authors correlate the increased CD45 expression with increased inflammation. This is not entirely accurate. Indeed, CD45 is abundantly expressed by Kupffer cells, so the increased CD45 expression could be related to the recruitment or proliferation of new Kupffer cells, which is not necessarily indicative of inflammation but also occurs during tissue regeneration. Moreover, the increased CD45 expression could result from an increased percentage of Kupffer cells compared to other non-parenchymal cells. Finally, hepatic non-immune cells, such as liver sinusoidal endothelial cells (LSECs), also express CD45.
To correlate the increased CD45 expression with recruitment of inflammatory cells, the authors should perform FACS analysis or PCR analysis on isolated liver immune cells rather than total liver lysates. Alternatively, the authors must rephrase and tone down their comments on this specific aspect.
Authors reply: We appreciate the reviewer for providing these innovative suggestions. The inflammatory reaction is not the main focus of the current study. Unfortunately, we are unable to repeat the animal experiments to conduct FACS analysis or PCR analysis on isolated liver immune cells in a timely manner for revision without a new ethics vote for additional animal experiments.
Based on the immunohistologic analysis of CD45 in our mice, we have determined that it is highly unlikely that CD45 expression in LSECs accounts for a significant proportion of hepatic CD45 expression (Fig. A, please refer to the attached pdf-file).
Fig. A: CD45 immunostaining visualized a significant hepatic infiltration of CD45+ leukocytes, particularly in the granulomas (arrows) of S. mansoni-infected animals, but also within the parenchyma (arrowheads). Con healthy control, Sm S. mansoni-infected, p: portal field, asterisk: S. mansoni egg, Scale bars: 100µm.
Regarding the reviewers´ comments on CD45, we have made alterations to the following chapter of the discussion (new additions are marked in red):
´CD45 is expressed on all nucleated hematopoietic cells and plays a central role in antigen receptor signal transduction and lymphocyte development [46]. Hepatic gene expression of the pan-leukocyte marker Cd45 increased after infection but it was not influenced by the host´s age at the time of infection (Fig. 3B). As the hepatic level of Cd45 ismay be used as a measure of inflammatory infiltration of the liver, it appears likely that, apart from granulomas, smaller accumulations of leukocytes within the parenchyma contribute to hepatocellular damage´.
Finally, it is well-known that liver fibrosis has a profound impact on T cell activation, and consequently, the different inflammatory reactions could be a secondary effect of the fibrosis, which reduces the ability of immune cells to perform their effector functions. For instance, liver fibrosis is characterized by a drastic reduction in endothelial porosity, which is one of the major reasons for reduced intrahepatic immune reactions (PMID: 26188075, 19033667). This aspect should at least be mentioned in the discussion and/or introduction.
Authors reply: We agree with the reviewer and have introduced the following sentences into the discussion:
´It is well-known that liver fibrosis has a profound impact on T cell activation. Consequently, the different inflammatory reactions could be a secondary effect of fibrosis, which reduces the ability of immune cells to perform their effector functions. For in-stance, liver fibrosis is characterized by a drastic reduction in endothelial porosity, which is one of the major reasons for reduced intrahepatic immune reactions [56,57].´

Reviewer 2 Report
Comments and Suggestions for Authors
This article presents the pathophysiology of Schistosoma infection clearly and concisely. Also, the methodology used is appropriate, and the results are based on the specialized literature. The discussions are elaborated, with topical references. I suggest that it is necessary to improve the final elements of the conclusion, these being presented briefly. The limitations of this article need to be revised.
Author Response
Response to Reviewer 2
This article presents the pathophysiology of Schistosoma infection clearly and concisely. Also, the methodology used is appropriate, and the results are based on the specialized literature. The discussions are elaborated, with topical references. I suggest that it is necessary to improve the final elements of the conclusion, these being presented briefly. The limitations of this article need to be revised.
Authors reply: We would like to thank the reviewer for his/her feedback. We have made improvements to the final elements of the conclusion and revised the limitations of this article based on the reviewers´ suggestions.

Reviewer 3 Report
Comments and Suggestions for Authors
Dear Author, Thank you for your work on this important issue.
Overall, the manuscript is well written, and the experiments are properly design however, to prove the age-dependent effect of parasite egg burden on liver fibrosis author should consider including:
(1) Proteomics and transcriptomics may reveal pathways or molecules that contribute to granuloma formation, the immune response, and fibrosis. Although useful, these markers are standard for all types of inflammation. Deeper examination of alternative pathways could lead to new therapeutic targets for treating liver fibrosis caused by parasitic infection.
(2) Author discussed metabolomics shift – hepatic carbohydreate metabolism by S. mansoni eggs and measured glycogen and hepatic mRNA levels of 3 genes: Hk2, Pkm2, and G6PD. The gene level does not always correspond with protein level therefore to prove the metabolic shift author should consider adding additional metabolic markers using Western blotting for glycolysis markers (e.g., GLUT1, LDHA), and analyze liver energy stores (glycogen) via PAS staining. Furthermore, employing metabolic profiling by metabolomics may shed some light on why the pentose phosphate pathway was reduced, as presented in Supplemental Fig. 16.
(3) Author measured only CD45, marker which is used as a marker for white blood cells and other immune cells. Profiling immune cells infiltrating the liver by staining for surface-specific markers such as CD11b (myeloid cells), F4/80 (macrophages), CD3 (T cells), Ly6G (neutrophils), M1 macrophages (iNOS, CD86) and M2 macrophages (Arg1, CD206) can provide detailed information on pathological pathways, particularly in younger subjects. Juanjuan Zhao et al. identified more than 30 discrete cell populations comprising 13 T and NK cells, 7 B cells, 4 plasma cells, and 8 myeloid cell subsets in the human liver and donor-paired spleen and blood and characterized their tissue distribution, gene expression and functional modules [https://doi.org/10.1038/s41421-020-0157-z]. Looking at the distribution of the different cells in age-dependent infection should be informative.
(4) Author could consider fibrosis modulation e.g., TGF-β inhibitors, IL-6 blockers, or macrophage polarization intervention to assess whether blocking key signaling pathways modulates fibrosis and granuloma formation.
Minor.
1. ALT – abbreviations should be explained when used first time.
2. How many human samples were included in the study. According to the text, only one human biopsy was used: “visualized in granuloma of a histologic slice of a liver biopsy that was routinely taken for diagnosis from a 31-year-old male patient suffering from schistosomiasis (SFig. 15)”.
3. The limitations pointed out by the author are quite substantial, as a lower number of eggs might reflect a lower number of adult worms, not necessarily an age-related decrease in egg laying capacity or disease severity. To rule out the limitation author should correlate the worm count with egg loads in the liver, and correlated ALT and fibrotic markers with both worm count and egg burdens for more comprehensive assessment. Author could check the following reference for more information [https://doi.org/10.1038/s41598-020-72901-y].
Author Response
Response to Reviewer 3
Dear Author, Thank you for your work on this important issue.
Overall, the manuscript is well written, and the experiments are properly design however, to prove the age-dependent effect of parasite egg burden on liver fibrosis author should consider including:
(1) Proteomics and transcriptomics may reveal pathways or molecules that contribute to granuloma formation, the immune response, and fibrosis. Although useful, these markers are standard for all types of inflammation. Deeper examination of alternative pathways could lead to new therapeutic targets for treating liver fibrosis caused by parasitic infection.
Authors reply: This is a pioneering suggestion and we will take up this idea in a follow-up study.
(2) Author discussed metabolomics shift – hepatic carbohydreate metabolism by S. mansoni eggs and measured glycogen and hepatic mRNA levels of 3 genes: Hk2, Pkm2, and G6PD. The gene level does not always correspond with protein level therefore to prove the metabolic shift author should consider adding additional metabolic markers using Western blotting for glycolysis markers (e.g., GLUT1, LDHA), and analyze liver energy stores (glycogen) via PAS staining. Furthermore, employing metabolic profiling by metabolomics may shed some light on why the pentose phosphate pathway was reduced, as presented in Supplemental Fig. 16.
Authors reply: This is a constructive suggestion. Nevertheless, we have already demonstrated before that the hepatic protein levels of glycolysis markers were induced in animals infected with S. mansoni (JHEP Rep. 2022;5(2):100625. doi: 10.1016/j.jhepr.2022.100625). Instead of qualitative PAS staining, we have already demonstrated the quantitative assessment of hepatic glycogen which is the most intriguing sign to prove the metabolic shift provoked by the eggs in the hosts liver as we demonstrated recently (JHEP Rep. 2022;5(2):100625. doi: 10.1016/j.jhepr.2022.100625).
We also thank you for the promising suggestion to employ metabolic profiling through metabolomics to shed light on why the pentose phosphate pathway was reduced. However, these investigations go far beyond the focus of the present study and are planned to be addressed in a follow-up study.
(3) Author measured only CD45, marker which is used as a marker for white blood cells and other immune cells. Profiling immune cells infiltrating the liver by staining for surface-specific markers such as CD11b (myeloid cells), F4/80 (macrophages), CD3 (T cells), Ly6G (neutrophils), M1 macrophages (iNOS, CD86) and M2 macrophages (Arg1, CD206) can provide detailed information on pathological pathways, particularly in younger subjects. Juanjuan Zhao et al. identified more than 30 discrete cell populations comprising 13 T and NK cells, 7 B cells, 4 plasma cells, and 8 myeloid cell subsets in the human liver and donor-paired spleen and blood and characterized their tissue distribution, gene expression and functional modules [https://doi.org/10.1038/s41421-020-0157-z]. Looking at the distribution of the different cells in age-dependent infection should be informative.
Authors reply: The reviewers’ suggestions offer a variety of novel approaches for follow-up studies. It seems promising to investigate the distribution of different cells in age-dependent S. mansoni infection, although this would be beyond the scope of our current study.
However, we also performed immunohistologic analysis of CD45 to characterize hepatic infiltration by immune cells (Fig. A, please refer to the attached pdf-file).
Fig. A: CD45 immunostaining visualized a significant hepatic infiltration of CD45+ leukocytes, particularly in the granulomas (arrows) of S. mansoni-infected animals, but also within the parenchyma (arrowheads). Con healthy control, Sm S. mansoni-infected, p: portal field, asterisk: S. mansoni egg, Scale bars: 100µm.
(4) Author could consider fibrosis modulation e.g., TGF-β inhibitors, IL-6 blockers, or macrophage polarization intervention to assess whether blocking key signaling pathways modulates fibrosis and granuloma formation.
Authors reply: We would like to thank the reviewer once again for these fantastic ideas that go beyond the central idea of the current study. The suggested animal experiments would require an ethics vote, and the realization of the projects would take months, so we have decided to use them in follow-up studies.
Minor.
- ALT – abbreviations should be explained when used first time.
Authors reply: We apologize for any inconvenience caused and would like to introduce alanine transaminase (ALT) when it is first mentioned.
- How many human samples were included in the study. According to the text, only one human biopsy was used: “visualized in granuloma of a histologic slice of a liver biopsy that was routinely taken for diagnosis from a 31-year-old male patient suffering from schistosomiasis (SFig. 15)”.
Authors reply: Yes, one human biopsy was used.
- The limitations pointed out by the author are quite substantial, as a lower number of eggs might reflect a lower number of adult worms, not necessarily an age-related decrease in egg laying capacity or disease severity. To rule out the limitation author should correlate the worm count with egg loads in the liver, and correlated ALT and fibrotic markers with both worm count and egg burdens for more comprehensive assessment. Author could check the following reference for more information [https://doi.org/10.1038/s41598-020-72901-y].
Authors reply: As mentioned in the limitations-chapter ´We refrained from recovering adult parasites by perfusion as a stable burden has been described in adult mice [14].´ However, we are grateful for the suggested reference that we cited in the materials and methods section to enhance the description of the assessment of hepatic egg load.

Round 2
Reviewer 3 Report
Comments and Suggestions for Authors
The major observed shortcoming in revised manuscripts is validation set, which is based on one human sample. Consequently, the validation set is too small and cannot validate the any data. I suggest removing it and modify the text adequate.
Author Response
Response to Reviewer 3:
The major observed shortcoming in revised manuscripts is validation set, which is based on one human sample. Consequently, the validation set is too small and cannot validate the any data. I suggest removing it and modify the text adequate.
Author´s reply: We thank the reviewer and apologize for the lack of an appropriate number of human samples for validation. These samples are very rare and we are constantly working to increase the number of samples in the validation set. We will definitely address this issue in future publications. However, we believe that validation with at least one human specimen should be shown if available. Despite the small validation set, we have left Fig. S15 in the revised manuscript and included the following sentence to line 538 of the revised manuscript to emphasize this point: “The availability of human samples for validation is limited and the presence of desmin+ cells in granulomas of human patients should be confirmed in future studies with additional material.´
